# Role of Cr Element in Highly Dense Passivation of Fe-Based Amorphous Alloy

**DOI:** 10.3390/ma16206630

**Published:** 2023-10-10

**Authors:** Ziqi Song, Zhaoxuan Wang, Qi Chen, Zhigang Qi, Ki Buem Kim, Weimin Wang

**Affiliations:** 1Key Laboratory for Liquid-Solid Structural Evolution and Processing of Materials (Ministry of Education), School of Materials Science and Engineering, Shandong University, Jinan 250061, China; 202234218@mail.sdu.edu.cn (Z.S.); 202114121@mail.sdu.edu.cn (Z.W.); 202120538@mail.sdu.edu.cn (Q.C.); 202034203@mail.sdu.edu.cn (Z.Q.); 2Department of Nanotechnology and Advanced Materials Engineering, Sejong University, Seoul 05006, Republic of Korea; kbkim@sejong.ac.kr

**Keywords:** Cr element, passivation, Fe based, amorphous alloys

## Abstract

The effect of the Cr element on the corrosion behavior of as-spun Fe_72−*x*_Cr*_x_*B_19.2_Si_4.8_Nb_4_ ribbons with *x* = 0, 7.2, 21.6, and 36 in 3.5% NaCl solution were investigated in this work. The results show that the glass formability of the alloys can be increased as Cr content (*c*_Cr_) is added up to 21.6 at.%. When *c*_Cr_ reaches 36 at.%, some nanocrystals appear in the as-spun ribbon. With increasing *c*_Cr_ content, the corrosion resistances of as-spun Fe-based ribbons are continually improved as well as their hardness properties; during the polarization test, their passive film shows an increase first and then a decrease, with the highest pitting potential as *c*_Cr_ = 7.2 at.%, which is confirmed by an XPS test. The dense passivation film, composed of Cr_2_O_3_ and [CrO*_x_*(OH)_3−2*x*_, *n*H_2_O], can reduce the number of corrosion pits on the sample surface due to chloride corrosion and possibly be deteriorated by the overdosed CrFeB phase. This work can help us to design and prepare the highly corrosion-resistant Fe-based alloys.

## 1. Introduction

Since firstly fabricated in the last century, the amorphous alloys have a wide range of applications in many industrial fields, due to their special atomic structures, high strengths, and high hardness properties; Fe-based metallic amorphous alloys have especially excellent soft magnetic properties [1]. Meanwhile, Fe-based amorphous alloys are also used in various fields due to their excellent corrosion/wear resistances, such as drill bits [2] and artificial bone joints [3]. In addition, Fe-based metallic glasses contain major components with much lower costs than Co, Zr, Ni, Cu, and Ti [4]. Nowadays, in the rapid and high-level development of industry, it is necessary to solve the problems of corrosion and wear of metallic parts working in harsh environments to increase the corrosion resistances of Fe-based amorphous alloys is in line with the current industrial needs [5,6,7].

In the known stainless steels, Cr is one of the important and cheap elements to improve the corrosion resistance of the alloys [8]. Adding Cr can improve the corrosion/wear resistance of Fe-based amorphous alloys drastically [9,10,11,12]. At high temperatures, a high Cr content can ensure the mechanical strength and corrosion resistance of the alloys simultaneously, so a high-Cr alloy with high temperature corrosion resistance is widely used in the manufacturing of turbochargers, furnace fixtures, etc. [13,14]. The corrosion resistance performance of Fe-based amorphous alloys is attributed to three main factors: (1) amorphous alloys have a uniform passivation film [15]; (2) they have a high reactivity to diffuse the passive components to the surface [16]; and (3) they have no crystal defects. In biomedicine, the Cr-contained Fe-based amorphous alloys have excellent corrosion resistance and a superior biocompatibility, suitable for preparing surgical scissors. The (Fe, Cr)-based glass exhibits better corrosion resistance than 316L steel and Ti-based commercial alloys in the human body-like environment [17,18]. The Fe-based amorphous alloys with Cr show high corrosion resistance in Cl^−^ environments, giving new ideas for the design of marine anti-corrosion alloys [2,19,20].

Mo is another important element to be often added to steel and cast iron to improve their corrosion resistance. According to the argument on the pitting resistance equivalent number (also denoted as PREM), the element Mo can increase the resistance to local pit corrosion of the alloy [21,22]. In the real preparation and machining process, some Mo is added to most stainless steels to improve their resistances to pitting corrosion. Generally, Mo has a high cost; other low cost elements are substituted and then reduce the cost of the alloys, keeping enough corrosion resistance [23]. In the alloys containing both Cr and Mo, Mo has a strong interaction with Cr to increase corrosion resistance, forming a Cr-rich passive oxide film, rather than forming a Mo-oxide film [24]. Thus, increasing the *c*_Cr_, replacing Mo with other low-cost corrosion resistance elements in the alloy, is a practical idea [25]. It is necessary to be pointed out that Fe-based amorphous alloys can have a stable passivation rather than the active dissoluble film only when the Cr content is beyond a certain critical value [26]. Therefore, it is valuable to explore the role of the composition in the mechanism of formation and destruction of passivation films of Fe-based amorphous alloys in this work.

The marine environment contains lot of Cl^−^ and serious natural corrosion conditions. In a series of industrial circumstances, such as off-shore oil/gas exploitation and underwater operations, the selection of alloy types with high corrosion resistances is an important approach to solve the Cl^−^-induced degradation of parts [27]. Recently, the effect of Cr mainly involves low Cr steel in case of CO_2_ corrosion [28]; however, its anti-corrosion mechanism in different solutions is different. In this paper, we mainly studied the corrosion behavior of Fe-based amorphous alloys with Cr in a simulated marine environment, i.e., Cl^−^-contained solution. At the same time, Mo in traditional Fe-based amorphous alloys containing Cr is replaced by Nb, Si, and other elements to improve the corrosion resistance and GFA of amorphous alloys. We discuss the effect of Cr on the corrosion behaviors of Fe-based amorphous alloys in neutral solutions. Meanwhile, the composition and destruction process of amorphous passivated films during electrochemical tests have been analyzed. The results of this work not only provide a new way to understand the effect of Cr on the corrosion resistance of Fe-based metallic glasses but also provide a new method for supplying the needed passivation film of Fe-based amorphous alloys.

## 2. Experimental

### 2.1. Sample Preparation

The alloy ingots with nominal compositions of Fe_72−*x*_Cr*_x_*B_19.2_Si_4.8_Nb_4_ (*x* = 0, 7.2, 21.6, and 36) were smelted with pure Fe (99.99 wt.%), pure Si (99.999 wt.%), pure Nb (99.99 wt.%), pure Cr (99.99 wt.%), and Fe-B ingots in an arc melting furnace (MAM-1 Edmund Buhler, Berlin, Germany) protected by high purity argon (99.999%). Each prepared ingot was 5 g and remelted at least four times during the smelting process to ensure the uniformity of the composition. The obtained ingots were made into amorphous ribbons at a roller speed of 41.8 m·s^−1^ under the protection of purified argon (99.999%) by a single roll melt spinning system (SD500 SKY, Shenyang, China).

### 2.2. Structural Analysis

The microstructures of the ingots and the as-spun ribbons as well as the ground ribbon powders were characterized by X-ray diffraction (XRD, Bruker D8 Discover, Beijing Technology Co., Ltd., Beijing, China). The thermodynamic behaviors of the ribbons were measured by differential scanning calorimetry (DSC, NETZSCH-404, Netzsch-404, Netzsch, Bavaria, Germany) under the protection of high purity argon at a heating rate of 20 K/min. The ribbon microstructures were observed by transmission electron microscopy (TEM, JEM-2100F, Japan Electronics Co., Ltd., Beijing, China), and the surface morphologies and passivation film compositions were analyzed by a scanning electron microscope (SEM, JSM-7800 F Japan Electronics Co., Ltd., Beijing, China) and X-ray photoelectron spectroscopy (XPS, AXIS Supra, Manchester, UK). 

### 2.3. Electrochemical Tests

An electrochemical workstation (CHI660E, Shanghai Chenhua Instrument Co., Ltd., Shanghai, China) was used to test and characterize the corrosion behavior of the ribbons at room temperature (300 K). The polarization curve and electrochemical impedance spectroscopy (EIS) of the ribbons were tested in 0.6 M NaCl neutral solution with a three-electrode system. One side of the ribbons was covered with silicone rubber 12 h in advance before the test to ensure the accuracy of the experiment. In addition, the ribbons were immersed in hydrochloric acid solution at pH = 3 for 215 h to calculate the weight loss rate, which was calculated by the following formula:(1)$$v_{weight}=\frac{m_{0}-m_{1}}{S\times t}$$

$m_{0}\;(g)$ is the weight of the ribbon before immersion, $m_{1}\;(g)$ is the weight of the ribbon after immersion, $S\;({cm}^{2})$ is the immersion area of the ribbon, and t (h) is the immersion time.

## 3. Results and Discussion

### Microstructure

Figure 1a–d shows the SEM images of the Fe_72−*x*_Cr*_x_*B_19.2_Si_4.8_Nb_4_ ingots with *x* = 0, 7.2, 21.6, and 36, denoted by Cr0, Cr7, Cr21, and Cr36, respectively. The corresponding XRD patterns are shown in Figure 1e–h. Based on the XRD patterns, the four ingots had a certain similarity; there were *α-*Fe phases. With the addition of Cr, three strong peaks of *α-*Fe gradually shifted to the left, indicating that the average atomic distance increased, probably because Cr atoms with larger atomic radii replaced some of the Fe atoms. In addition, with the addition of the Cr element, CrFeB phases appeared in the ingots. Interestingly, from the ingots, the tendency of CrFeB phase dendritic growth was more and more evident. It was related to the migration and diffusion of solute during solidification, which was in line with the classical theory of “constitutional supercooling” proposed by Thiller [29]. In Cr0 ingot, the FeB phase presented in large plate-like shape (Figure 1a). With the addition of the Cr element, an elongated CrFeB phase began to appear in the ingot, indicating a strong binding force between Cr and Fe (Figure 1b,d). Some elliptical protrusions of Nb-Si intermetallic compounds in the ingots were formed, indicating that the strengths of Nb-Si bonds were stronger than those the of Nb-Nb bonds (Figure 1c,d).

The microstructures of the Cr0, Cr7, Cr21, and Cr36 as-spun ribbons and the as-crushed Cr36 ribbon were investigated by XRD. The diffraction patterns of the as-spun Cr0, Cr7, and Cr21 ribbons exhibited a typical diffuse peak within the range of 44.2 ± 0.4, without evident sharp crystallization peaks, indicating the amorphous properties of these samples (Figure 2a–c). This indicated that the FeSiBNb alloy with a small amount of Cr could form amorphous ribbons. As the Cr content increased, the position of the diffuse emission peak decreased from 44.6° to 43.2°, indicating an increase in the average atomic spacing, and the free volume within the ribbons also increased [30]. At the same time, the ribbons became increasingly brittle, and the XRD of the Cr36 ribbon showed evident crystal peaks. Cr36 ribbons became shorter and more brittle, which may have been related to the existence of internal stress during the ribbons production process. As shown in Figure 2d,e, there is a set of crystal peaks on the XRD curve of Cr36 ribbon; the *α*-Fe phase is not evident in the as-spun Cr36 ribbon, but the *α*-Fe peak becomes very evident after the sample has been crushed into pieces, similar to Ref. [31], indicating that the orientation of CrFeB and *α*-Fe are developed in the as-spun ribbon and that the *α-*Fe phase in the ingot also obtained a certain inheritance in the ribbons. These results suggested the structural heredity between the melts/glasses and ingots of present alloys.

In order to further characterize the microstructure of the as-spun Cr36 ribbon, TEM tests were performed. The TEM images and selected area electron diffraction (SAED) patterns of Cr36 ribbon are shown in Figure 3. They present many nanocrystals in the Cr36 ribbon. As shown in Figure 3b,c, the grain diameter is about 100 nm, and there are some dislocation structures in the rod-shaped grains, which are distributed vertically along the head and tail of the rod-shaped grains. At the same time, EDS shows that the rod-shaped grains have oxygen enrichment, indicating that higher activation energy is displayed at the dislocations, and the metal oxide reaction preferentially occurs in the regions with high dislocation density [32]. Figure 3(d_1_,d_2_) are SAED corresponding to HRTEM (d) (High Resolution Transmission Electron Microscope), corresponding to the CrFeB and *α*-Fe phases, respectively.

The DSC curves of the Cr0, Cr7, Cr21, and Cr36 ribbons with heating/cooling rates of 20 K/min are shown in Figure 4. The characteristic thermodynamic temperatures are listed in Table 1. The amorphous Cr0, Cr7, and Cr21 ribbons exhibit similar exothermic behaviors, and all the amorphous samples have only one exothermic crystallization peak, usually corresponding to eutectic crystallization [33]. The height of the crystallization peak increases with the increase in Cr content. As the Cr content increases up to 21.6 at.%, the reduced crystallization temperature *T*_rx_ (=*T*_x_/*T*_l_) and ∆*H*_rc_ of amorphous ribbons also gradually increases, indicating that the addition of 21.6 at.% Cr element can still improve the amorphous stability and amorphous formation ability of Fe-based ribbons [34]. When the addition of the Cr element reaches 36 at.%, there is no exothermic peak in the sample in Figure 4a, indicating that the ribbons has crystallized. In addition, the onset temperature of melting (*T*_m_) of the amorphous ribbons shows an increasing trend, and the end temperature of melting (*T*_l_) does not tends to decrease with increasing *x* (Figure 4b). In other words, the solidification temperature range Δ*T*_l_ (*T*_l_–*T*_m_) of ribbons does not change monotonically. The results show that Cr21 ribbon is the closest component to the eutectic point [30].

In general, the smaller the residual penetration depth, the higher the nanohardness. The steeper the unloading curve, the larger the Young’s modulus [35,36,37]. The nanoindentation images of Cr0, Cr7, Cr21, and Cr36 ribbons are shown in Figure 5, with the increase in Cr content, the hardness and Young’s modulus of the ribbons both increased. The hardness and Young’s modulus of the Cr36 ribbon as a crystal increase significantly, indicating that Cr-rich ordered clusters in the ribbon can improve the hardness of the ribbon.

Figure 6 shows the potentiodynamic polarization plots of the Cr0, Cr7, Cr21, and Cr36 as-spun ribbons and the corrosion rate *R*_im_ deduced from the weight loss in the HCl solution with a pH = 3. The corrosion potential *E*_corr_ (vs. SCE (saturated calomel (reference) electrode)), corrosion current density *i*_corr_ (μA·cm^−2^), pitting potential *E*_pit_ (V vs. SCE), and deduced corrosion rate CR (μm·y^−1^) with Equation (2) are summarized in Table 2 [38,39].
CR = 3.28 *i*_corr_ × *M/nd* (μm × y^−1^)(2)

Here, *M* is the atomic weight of Fe (55.85 g), *n* is the number of electrons transferred in the corrosion reaction (*n* = 2), and *d* is the density of Fe (7.88 g·cm^−3^). For the sake of similarity, the corrosion rate is mainly ascribed to the dissolution of Fe atoms.

From the potentiodynamic polarization curves (Figure 6a), the amorphous ribbon Cr0 has no evident passivation platform, the anode polarization curves of Cr7, Cr21, and Cr36 have significantly wide passivation platforms, and there is no evident anodic Tafel region. The corrosion current density was determined by cathodic polarization curves and corrosion potential. In addition, the addition of Cr increased the corrosion potential of the ribbons and decreased the corrosion current density of the ribbons (Table 2). As can be seen from Figure 6b, with the increase in Cr content, the corrosion rate of the ribbons after immersing in HCl solution with pH = 3 for 215 h decreased, which was consistent with the Tafel curves. At a high potential of about 0.7 V, Cr transpassive dissolution occurred in Cr36 ribbon, according to Ref. [40], resulting in a gradual increase in current. When the potential was less than 0, with the increase in *c*_Cr_ (Table 2), the value of *i*_corr_ and *E*_corr_ of the ribbons decreased, which was consistent with Figure 6b, indicating that the corrosion resistance of the ribbons was getting better. When the potential was increased until the ribbons were pitted, it can be seen that the *E*_pit_ of the ribbons reached the maximum value in Cr7, indicating that Cr7 had the best pitting corrosion resistance. On the one hand, the disorder of amorphous ribbons reduced the existence of crystal defects and prevented the formation of galvanic corrosion. On the other hand, the composition contained more easily passivated elements (such as Cr), which was conducive to improving corrosion resistance [41].

In comparison, we collect the date of 51 Fe-based glassy alloys, as shown in Figure 7. The *i*_corr_ and *E*_corr_ of Fe-based glassy alloys are listed in Appendix A. Apparently, Cr36 has a very low *i*_corr_ and a certain positive *E*_corr_, showing a super corrosion resistance.

Figure 8 shows the surface schematic graph morphologies and cross-sections of the tested samples after the polarization corrosion experiment in the neutral solution. In the case of Cr0, a corrosion scale in the thickness of 1–2 μm (Fe_2_O_3_ and SiO_2_) was mostly detached, with only small amounts remaining, and evident stress cracking was observed on the surface, exposing the matrix under the scale. In contrast, very little corrosion product was observed over the smooth surface of Cr7 (Figure 8b). It is thought that a relatively Cr-rich passive oxide may be produced on the surface, indicating that adding a small amount of Cr element can reduce the occurrence of local corrosion pits [54]. However, corrosion pits on the Cr21 ribbon’s surface are more evident than Cr7. At this time, small pits are easily formed on the bottom of the early formed large pit. On the surface of the Cr36 ribbon, it can be observed that the ribbon surface is rougher than Cr21 (Figure 8d), and the surface passivation film caused by pitting corrosion is in a chain-like shape. These results show that the addition of Cr element can firstly reduce stress cracking and pitting by corrosion and then increase them, show a maximal pitting resistance at *c*_Cr_ = 7 at.%.

To further understand the surface property changes, Figure 9 displays the XPS etching analysis on the surfaces of the Fe_72−*x*_Cr*_x_*B_19.2_Si_4.8_Nb_4_ ribbons with *x* = 0, 7.2, 21.6, and 36 after the potentiodynamic polarization in neutral solution. On the Fe 2p spectra (Figure 9(a_0_–a_3_)), the XPS curves of four ribbons can be decomposed into two peaks at about 707 and 709/710 eV, being identified as Fe^0^ and Fe^2+^/Fe^3+^, respectively [55]. As can be seen from Figure 8(a_0_,a_1_), the Fe^3+^ peak at 709.9 eV in Cr0 and Cr7 tend to decrease with the increase in etching time, whereas the Fe^3+^/Fe^2+^ peak at 709.3/710 eV in Cr21 and Cr36 gradually increases, which is because the passivation films of the Cr21 and Cr36 are thicker and denser. Under the same etching time, the passivation films of Cr36 are still Fe^2+^; there is no conversion of Fe^2+^ to Fe^3+^, which is consistent with the low corrosion current density and high corrosion potential shown in the Tafel curve (Figure 6a), indicating that a dense passivated film containing Fe is formed on its surface and that the Fe element finally has experienced interactive oxidation in the passivation film.

On most B 1s spectra (Figure 9(b_0_–b_3_)), we can decompose the XPS curves into two peaks at 192 and 188 eV, being indexed as B^3+^ and (Fe, Cr)B, respectively. It can be seen from Figure 9(b_0_–b_3_) that the oxides of the B element only exist on the surfaces of Cr7 and Cr21, and the contents of oxides are small; the (Fe, Cr)B compounds peak are stronger, and exist in Cr7, Cr21 and Cr36. Figure 9(b_0_–b_3_) shows that most of the element B oxides are corroded away during the corrosion process, indicating that element B is not the main factor in improving the corrosion resistance of the ribbons.

On the O 1s spectra, the XPS curves can be decomposed two peaks at about 531 and 532 eV, identified as OH^−^ (531.3 eV) and O^2−^ (532.9 eV), respectively [56]. With the increase in etching time, the reduction rate of OH^−^ of the four ribbons is greater than the reduction rate of O^2−^, indicating that some O^2−^ still exists at a certain depth of etching and passivation film composed of Fe_2_O_3_ and Cr_2_O_3_, and a certain amount of Si, Nb, and B element oxides is formed on the surface of the sample. Meanwhile, the oxide thickness of Cr7 and Cr21 is much higher than Cr0 and Cr36, due to their higher amorphous fraction *f*_a_ (Figure 4a).

On the Cr 2p spectra, we can divide the XPS curve into two peaks, identified as Cr^0^ (574.1 and 583.5 eV) and Cr^3+^ (577 and 586.4 eV), respectively, according to Refs. [57,58]. For Fe-based amorphous ribbons containing corrosion-resistant elements such as Cr, the surface of the amorphous ribbons will form a passivation film containing corrosion resistant elements, which will improve the corrosion resistance [59]. Cr is easy to form passivation films of chromium oxide and chromium hydroxide on the surfaces of amorphous ribbons, thus protecting the amorphous phase and inhibiting the active dissolution of the amorphous phase [60]. The formation mechanism can be explained by the following equation:2Cr + 3H_2_O = Cr_2_O_3_ + 6H^+^ + 6e^−^(3)

Then, part of the Cr oxide will further form H_2_O and Cr hydroxide [CrO*_x_*(OH)_3−2_*_x_*, *n*H_2_O]; both oxide films can improve the resistance of the ribbons to chloride ion corrosion [61].

As can be seen from Figure 9, Appendix A, the passivation film thicknesses of Cr7 and Cr21 ribbons are slightly larger than those of a Cr36 ribbon; among them, Cr21 has the best amorphous formation ability, so a thicker passive layer is formed on the surface. As can be seen from Figure 6a, Cr7 has a strong pitting corrosion resistance and is more stable, indicating that the oxide thickness is not the sole factor for its stability. However, it can be seen from the *i*_corr_ and *E*_corr_ of Cr36 in Table 2 that the corrosion resistance of Cr36 is still the best, indicating that a dense passivation film is formed on the surface of Cr36 when the potential is lower than 0 V.

Figure 10 shows the Nyquist and Bode plots with fitting results of Fe_72−*x*_Cr*_x_*B_19.2_Si_4.8_Nb_4_ glassy ribbons with *x* = 0, 7.2, 21.6, and 36 in a neutral solution. Among several equivalent circuit candidates, the equivalent circuit R(Q(R(CR))) is suitable for fitting the EIS data. The non-ideal capacitive behavior due to local inhomogeneity is represented by the constant phase element CPE (*Q*). The impedance of a CPE is defined as [62,63,64]:
(4)$$Q={\left(j\omega \right)}^{-n}/Y_{0}$$

Here, *Y*_0_ is the frequency independent parameter (Ω^−1^cm^−2^s^n^), *j* is the imaginary number, ω is the angular frequency (rad s^−1^), and *n* indicates the CPE power, which is between 0.5 and 1. When *n* = 1, the Q describes a pure capacitor. For 0.5 < *n* < 1, the Q represents a distribution of dielectric relaxation times in the frequency domain, and Q represents a Warburg impedance with diffusion character (*n* = 0.5).

The fitting results are summarized in Table 3. Here, the change transfer resistances (*R*_ct_) of the ribbons increase from 1.1 to 2410 kΩ·cm^2^ with the increase in *c*_Cr_, indicating that the passivation film becomes more stable and dense, which is consistent with Ref. [65]. Meanwhile, we can see that the value of *n* ranges from 0.83 to 0.88. The *n* value of the Cr0 ribbon is low, which indicates a higher dispersion at the alloy/electrolyte interface, probably due to the defects on the passive film formed in this hypersaline solution [66,67]. With increasing *c*_Cr_, the diameter of the Nyquist semicircle of Fe-based glassy ribbons significantly expands to the larger size (Figure 10a). The Bode plots is mainly divided into three different regions: (i) the low frequency region (10^−2^–1 Hz) mainly reflects the capacitance and resistance at the interface between the ribbons and the passivation film; (ii) the intermediate frequency region (1–10^3^ Hz) mainly reflects the resistance and capacitance of the passivation film; and (iii) the high frequency region (10^2^–10^5^ Hz) in this work mainly reflects the capacitances and resistances of the corrosion products [68,69,70]. In the bode plots, the closer the phase angle is to 90°, the more stable and compact the passivation film on the corrosion surface of the ribbons will be; it can be seen that the phase angle of the ribbons in the intermediate frequency region is closer to 90° when the element Cr is added [71]. Therefore, it can be seen that the increase in Cr content can make the passivation film more dense and thick, which is consistent with the passivation platform and the corrosion current density *i*_corr_ on the dynamic potential polarization curve (Figure 6a). The total resistance after fitting increases with the increase in Cr content, which is consistent with the Tafel results (Figure 6a).

## 4. Conclusions

In this work, we prepared Fe_72−*x*_Cr*_x_*B_19.2_Si_4.8_Nb_4_ ingots with *x* = 0, 7.2, 21.6, and 36 by arc melting, and ribbons with melt spinning in a vacuum. The microstructures of the ingots were investigated; microstructure investigations, hardness tests, electrochemical experiments, and XPS analyses on ribbons were conducted; and the surface morphologies of the ribbons after testing were analyzed. Combining the heredity between the ingots and ribbons, the effect of the Cr element on the properties of the Fe-based amorphous alloys could be found, as follows:(1)With increasing Cr content *c*_Cr_ (*x*), the stability and glass formability (GFA) of Fe-based alloys firstly increased until *x* = 21 and then decreased drastically as *x* = 36. Simultaneously, the CrFeB phase and clusters in the samples tended to form with strong orientations in the ingots and as-spun ribbons, which deteriorated the GFA of the alloy, as its abundance was high enough.(2)With increasing *c*_Cr_, the corrosion rate *R*_im_, corrosion current *i*_corr_, and deduced corrosion rate CR of Fe-based ribbons increased monotonically, as well as their hardness, whereas their pitting resistance increased until *x* = 7 and decreased as *x* = 21, indicating that the most stable passive film formed at a proper *c*_Cr_, which was confirmed by XPS data. This work can give a new clue to design and prepare highly corrosion-resistant Fe-based alloys.

## Figures and Tables

**Figure 1 materials-16-06630-f001:**
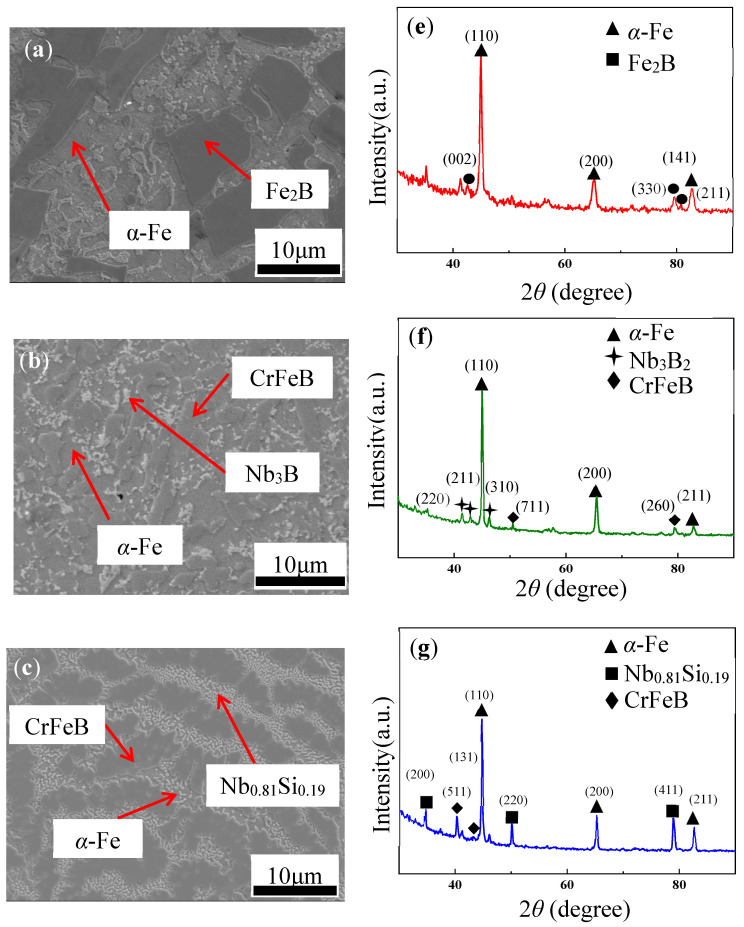
SEM micrographs of the Fe_72−*x*_Cr*_x_*B_19.2_Si_4.8_Nb_4._ (**a**) *x* = 0 (Cr0), (**b**) *x* = 7.2 (Cr7), (**c**) *x* = 21.6 (Cr21), and (**d**) *x* = 36 (Cr36) ingots and XRD patterns of (**e**) Cr0, (**f**) Cr7, (**g**) Cr21, and (**h**) Cr36.

**Figure 2 materials-16-06630-f002:**
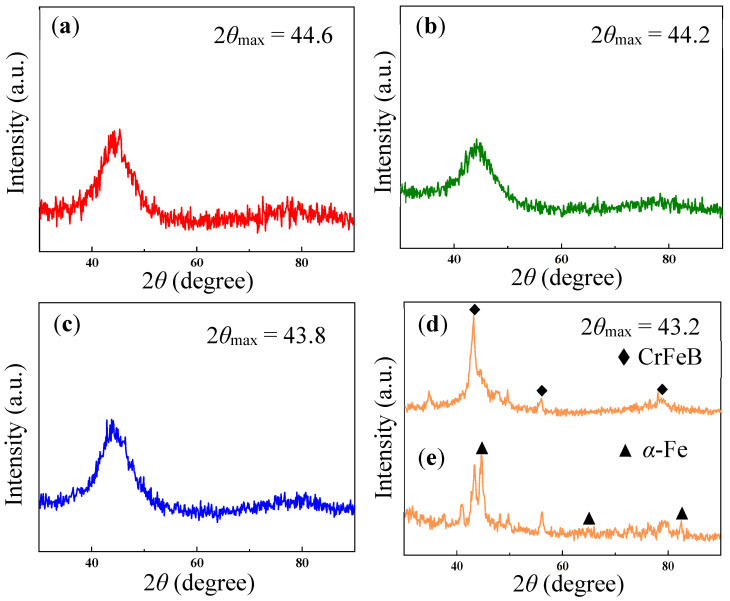
XRD curves of as-spun Fe_72−*x*_Cr*_x_*B_19.2_Si_4.8_Nb_4_ ribbons together with a crushed sample. (**a**) *x* = 0 (Cr0) ribbon, (**b**) *x* = 7.2 (Cr7) ribbon, (**c**) *x* = 21.6 (Cr21) ribbon, (**d**) *x* = 36 (Cr36) ribbon, and (**e**) crushed Cr36 sample.

**Figure 3 materials-16-06630-f003:**
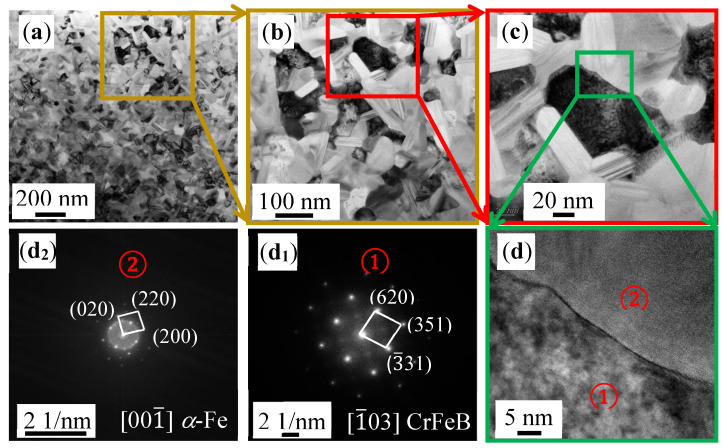
(**a**) LRTEM (Low Resolution Transmission Electron Microscope) image of Fe_36_Cr_36_B_19.2_Si_4.8_Nb_4_ (Cr36). (**b**,**c**) are the enlarged part of (**a**); (**d**) is the HRTEM (High Resolution Transmission Electron Microscope) image of Cr36 (① and ② represent different phases); and (**d_1_**) and (**d_2_**) are the corresponding FFT (Fast Fourier Transform) patterns.

**Figure 4 materials-16-06630-f004:**
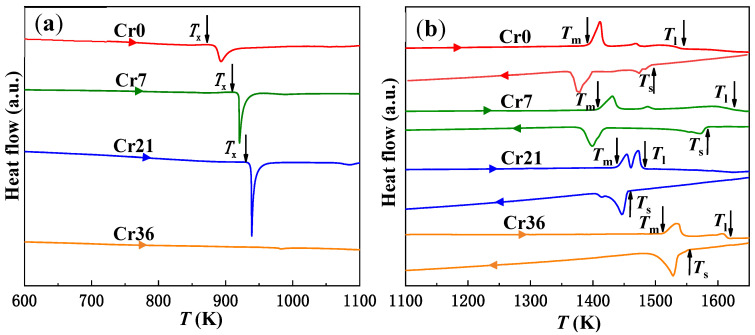
DSC curves of Fe_72−*x*_Cr*_x_*B_19.2_Si_4.8_Nb_4_ with *x* = 0 (Cr0), *x* = 7.2 (Cr7), *x* = 21.6 (Cr21), and *x* = 36 (Cr36) ribbons at different temperature intervals. (**a**) Heating curves in 600–1100 K with 20 K/min, and (**b**) heating and cooling curves in 1100–1650 K.

**Figure 5 materials-16-06630-f005:**
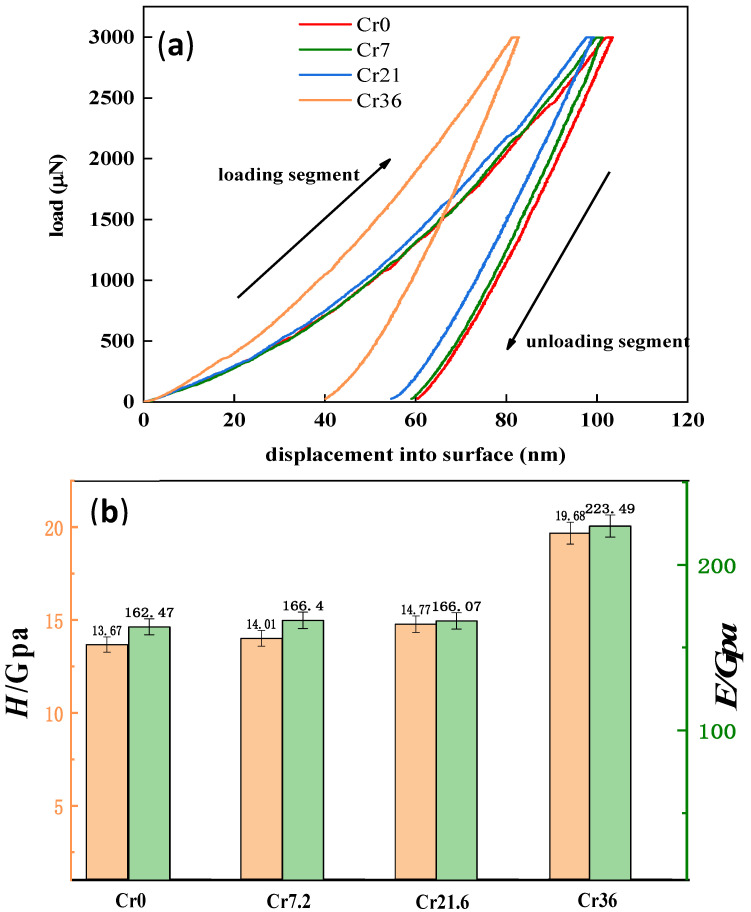
Nanoindentation of Fe_72−*x*_Cr*_x_*B_19.2_Si_4.8_Nb_4_ with *x* = 0 (Cr0), *x* = 7.2 (Cr7), *x* = 21.6 (Cr21), and *x* = 36 (Cr36) ribbons: (**a**) Curve of the load and indentation depth; (**b**) calculated results of the nanoindentation, including hardness and modulus.

**Figure 6 materials-16-06630-f006:**
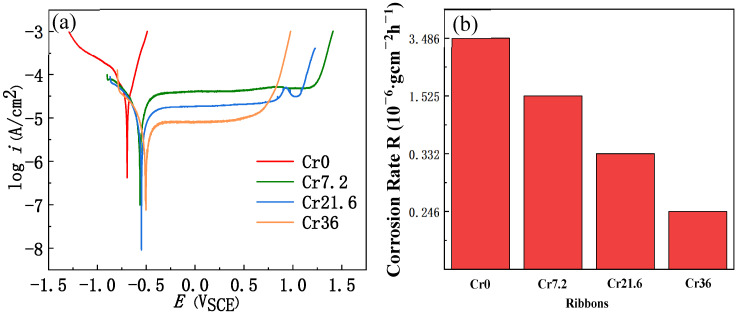
(**a**) Potentiodynamic polarization plots of as-spun ribbons Fe_72−*x*_Cr*_x_*B_19.2_Si_4.8_Nb_4_ with *x* = 0 (Cr0), *x* = 7.2 (Cr7), *x* = 21.6 (Cr21), and *x* = 36 (Cr36) in neutral solutions. (**b**) Corrosion rate *R*_im_ derived from weight loss obtained by immersing in acid solution.

**Figure 7 materials-16-06630-f007:**
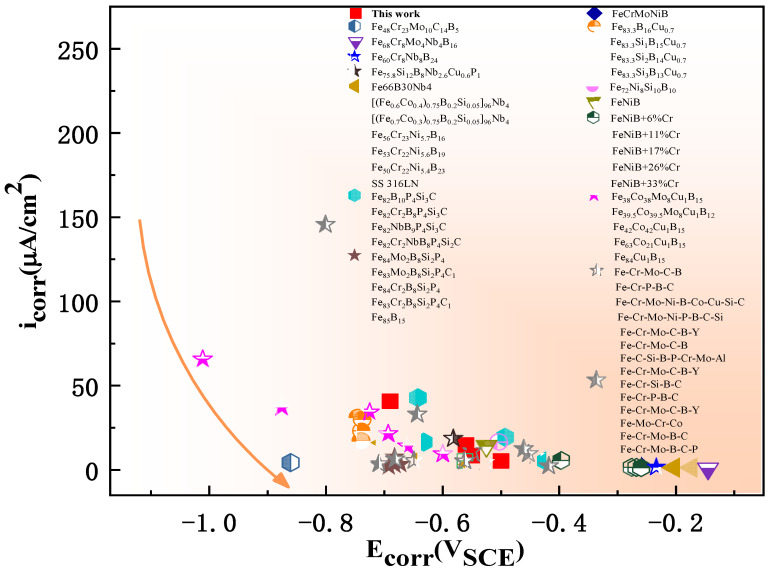
The comparison diagram of corrosion potential (*E*_corr_) and corrosion current density (*i*_corr_) of various Fe-based alloys in 3.5 wt.% NaCl solution; details are listed in Appendix A [42,43,44,45,46,47,48,49,50,51,52,53].

**Figure 8 materials-16-06630-f008:**
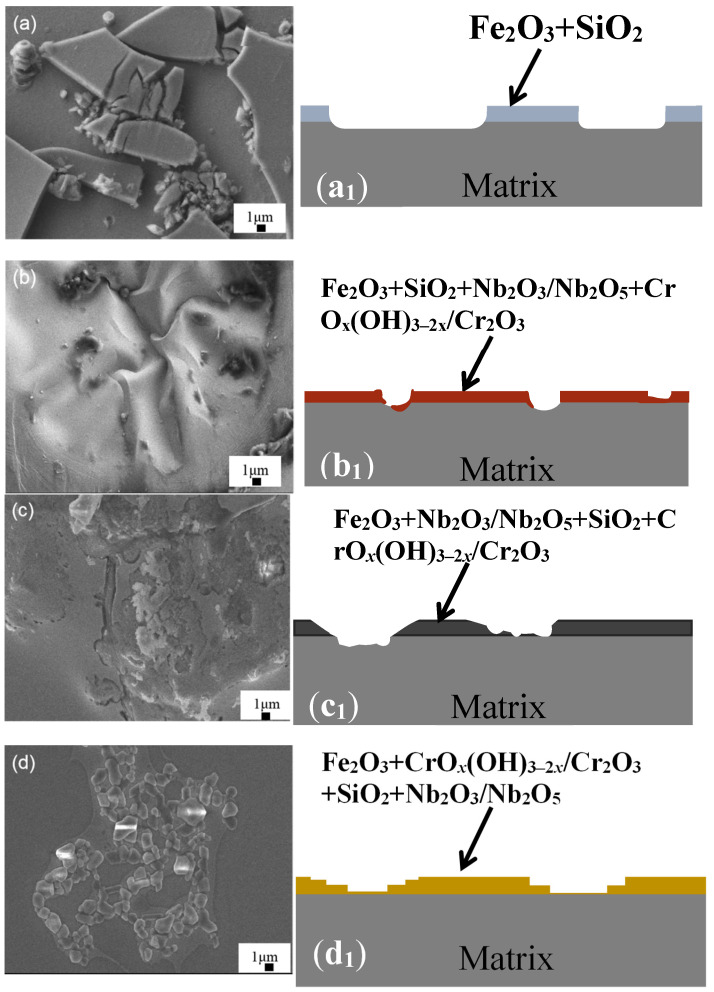
SEM images and simplified cross-sectional schematics of corroded Fe_72−*x*_Cr*_x_*B_19.2_Si_4.8_Nb_4_ samples. (**a**) *x* = 0 (Cr0). (**b**) *x* = 7.2 (Cr7). (**c**) *x* = 21.6 (Cr21). (**d**) *x* = 36 (Cr36). (**a_1_**–**d_1_**) Simplified cross-section schematics showing their damaged characteristics.

**Figure 9 materials-16-06630-f009:**
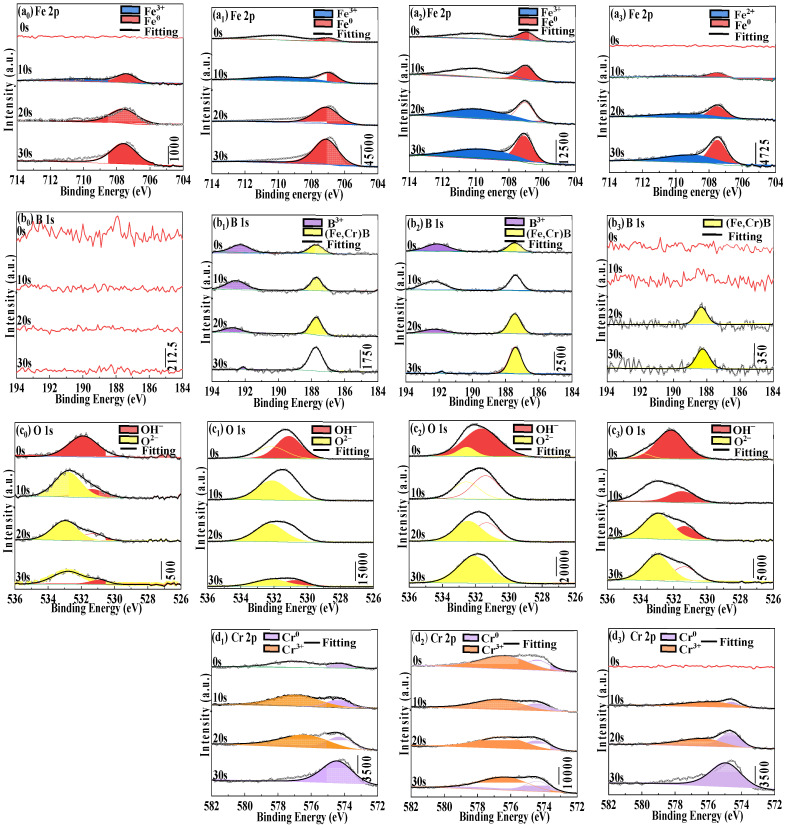
XPS spectra of Fe 2p (**a_0_**–**a_3_**), B 1s (**b_0_**–**b_3_**), O 1s (**c_0_**–**c_3_**) and Cr 2p (**d_1_**–**d_3_**) in binding energy regions for the as-spun Fe_72−*x*_Cr*_x_*B_19.2_Si_4.8_Nb_4_ ribbons with *x* = 0 (Cr0), *x* = 7.2 (Cr7), *x* = 21.6 (Cr21), and *x* = 36 (Cr36) after corrosion in neutral solutions.

**Figure 10 materials-16-06630-f010:**
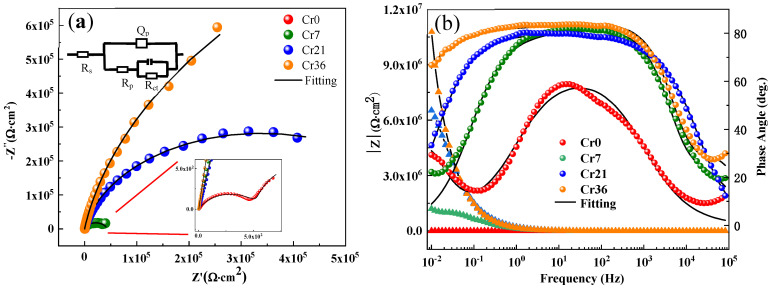
(**a**) Nyquist plots of as spun Fe_72−*x*_Cr*_x_*B_19.2_Si_4.8_Nb_4_ ribbons with *x* = 0 (Cr0), 7.2 (Cr7), 21.6 (Cr21), and 36 (Cr36) in neutral solution with equivalent circuits R(Q(R(CR))) and (**b**) the corresponding Bode plots. Symbols show the experimental data while solid lines are fitting results.

**Table 1 materials-16-06630-t001:** Thermal parameters deduced from the DSC curves, including the onset crystallization temperature *T*_x_, the solidification temperature *T*_s_, the onset melting temperature *T*_m_, the liquidus temperature *T*_l_, the solidification temperature range ∆*T*_l_ (=*T*_l_–*T*_m_), the reduced crystallization *T*_rx_ (=*T*_x_/*T*_l_), the degree of supercooling ∆*T* (=*T*_l_–*T*_s_), and the reduced crystallization heat Δ*H*_rc_ (=Δ*H*_c_/Δ*H*_m_). Here, Δ*H*_c_ and Δ*H*_m_ are heats of crystallization and melting, respectively.

Ribbon	*T*_x_ (K)	*T*_s_ (K)	*T*_m_ (K)	*T*_l_ (K)	Δ*T*_l_ (K)	Δ*T* (K)	*T* _rx_	Δ*H*_rc_ (Δ*H*_c_/Δ*H*_m_)
*x* = 0	882	1492	1394	1541	147	49	0.57	0.18
*x* = 7.2	916	1580	1406	1619	213	39	0.57	0.23
*x* = 21.6	933	1459	1438	1482	44	23	0.63	0.52
*x* = 36	-	1554	1512	1615	103	61	-	-

**Table 2 materials-16-06630-t002:** Parameters deduced from potentiodynamic polarization plots in Figure 6, such as corrosion potential *E*_corr_ (V vs. SCE), corrosion current density *i*_corr_ (μA·cm^−2^), pitting potential *E*_pit_ (V vs. SCE), and deduced corrosion rate CR (μm·y^−1^).

Ribbon	*E*_corr_ (V)	*i*_corr_ (μA·cm^−2^)	*E*_pit_ (V)	CR (μm·y^−1^)
*x* = 0	−0.69	40.72	−0.60	473
*x* = 7.2	−0.56	14.83	1.21	172
*x* = 21.6	−0.55	8.43	1.04	98
*x* = 36	−0.50	5.28	0.68	61

**Table 3 materials-16-06630-t003:** Simulated parameters of the Fe_72−*x*_Cr*_x_*B_19.2_Si_4.8_Nb_4_ ribbons with *x* = 0, 7.2, 21.6, and 36, derived from EIS curves. *R*_s_: the solution resistance, *Q*_p_: the passive film capacitance, *R*_p_: the passive film resistance, and *R*_ct_: the charge transfer resistance.

Ribbon	*R*_s_(Ω·cm^2^)	*Q* _p_		*R*_p_ (kΩ·cm^2^)	*C* (μF·cm^2^)	*R*_ct_ (kΩ·cm^2^)	*R*_total_ (kΩ·cm^2^)
		*Y*(10^−5^ Ω^−1^s^n^cm^−2^)	*n*				
*x* = 0	3.5	50.6	0.72	0.5	31,790	1.1	1.7
*x* = 7.2	1.1	3.3	0.84	0.0	9.8	43.0	43.0
*x* = 21.6	3.0	1.2	0.88	0.1	0.4	678.6	678.7
*x* = 36	1.3	1.1	0.86	3.2 × 10^−3^	6.0	2410	2410

## Data Availability

Data available on request due to restrictions eg privacy or ethical. The data presented in this study are available on request from the corresponding author. The data are not publicly available due to follow-up research on the work.

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
