# Peer review of "Role of Cr Element in Highly Dense Passivation of Fe-Based Amorphous Alloy"

_materials, 2023, doi:10.3390/ma16206630_

Round 1
Reviewer 1 Report
1. Similarity index is more than 35%, especially the abstract, introduction, experimental must be revised to reduce the plagiarism.
2. Introduction lacks the recent literature and motivation to perform this research. It has serious amount of similarity index.
3. Revise the sentece in a line 80 to 83 for more clarity.
4. Quality of the figures are poor and authors must add high resolution images.
5. During polarization test, the passive film depicted a significant increase and followed by decrease with the pitting potential. What is the reason for this?
6. Some of the unidentified diffraction peaks are present in figure 1 (e), (f) (h) and 2 (d), (e). Are these peaks are impurity peaks?
7. In DSC curve the gap between Tm and Tl is less for Cr21 sample. What is reason for that?
8. Confirm whether the figure 7 is original or adopted from elsewhere. If it is adopted, it is recommended to submit the copyright.
A minor revision is required for English
Author Response
Response to the comments on “Role of Cr element in highly dense passivation of Fe-based amorphous alloy”
- Similarity index is more than 35%, especially the abstract, introduction, experimental must be revised to reduce the plagiarism.
Re:
We are very sorry that the similarity index is more than 35%. According to the reviewer's advice, we re-read the literature and carefully sorted out the abstract, introduction and experimental parts, which is marked with red underlines.
- Introduction lacks the recent literature and motivation to perform this research. It has serious amount of similarity index.
Re:
We agree with the reviewer's advice. We have read relevant literature and made modifications to the “Introduction” part [1-5], and have added a description of the research significance and motivation of this work, which is marked with red underlines (Page 2, line 70-77 in the revised manuscript).
- Revise the sentece in a line 80 to 83 for more clarity.
Re:
Thanks for your comments, we have revised the sentence on lines 80 to 83 to make it more clarity. (Page 2, line 95-97 in the revised manuscript).
- Quality of the figures are poor and authors must add high resolution images.
Re:
We are very sorry for this problem and we have changed the images in the manuscript to one with higher resolution.
- During polarization test, the passive film depicted a significant increase and followed by decrease with the pitting potential. What is the reason for this?
Re:
It can be seen from the polarization curves that there is no passivation platform in Cr0 ribbon (x = 0) due to the absence of anti-corrosion element Cr. Cr7 and Cr21 ribbons (x = 7.2 and 21.6) have a higher Epit, because they have Cr element and are in fully amorphous state. However, when cCr reaches 36 at.%, the ribbon have nanocrystalline precipitates in the amorphous matrix, which deteriorate the stability of the passive film and has a lower Epit.
- Some of the unidentified diffraction peaks are present in figure 1 (e), (f),(h) and 2 (d), (e). Are these peaks are impurity peaks?
Re:
Some peaks in Figure 1 (e), (f), (h) are not recognized in Jade, possibly due to the metastable phase that occurs during rapid cooling process. In Figure 2 (d) and (e), some nanocrystals were doped in amorphous, so some crystallization peaks Jade could not be calibrated.
- In DSC curve the gap between Tmand Tl is less for Cr21 sample. What is reason for that?
Re:
We attribute the less gap between Tm and Tl of Cr21 sample to the fact that the composition is closest to the eutectic component, and the initial reaction is inhibited when the cCr (at%) reaches 21%.
- Confirm whether the figure 7 is original or adopted from elsewhere. If it is adopted, it is recommended to submit the copyright.
Re:
Figure 7 is the original figure. Ecorr and icorr of some Fe-base alloys were summarized by referring to literature, and all data and inserted literature were listed in table S1.
|
Sample |
Ecorr (V) |
icorr (mA cm-2) |
|
|
Cr0 |
-0.69 |
40.72 |
|
|
Cr7 |
-0.56 |
14.83 |
|
|
Cr21 |
-0.55 |
8.43 |
|
|
Cr36 |
-0.50 |
5.28 |
|
|
Fe66B30Nb4 |
-0.7 |
15 |
[6] |
|
[(Fe0.6Co0.4)0.75B0.2Si0.05]96Nb4 |
-0.55 |
2 |
|
|
[(Fe0.7Co0.3)0.75B0.2Si0.05]96Nb4 |
-0.63 |
7 |
|
|
Fe56Cr23Ni5.7B16 |
-0.13 |
0.015 |
|
|
Fe53Cr22Ni5.6B19 |
-0.2 |
0.06 |
|
|
Fe50Cr22Ni5.4B23 |
-0.17 |
0.02 |
|
|
SS 316LN |
-0.2 |
0.2 |
|
|
Fe82B10P4Si3C |
-0.64 |
41.7 |
[7] |
|
Fe82Cr2B8P4Si3C |
-0.63 |
15.3 |
|
|
Fe82NbB9P4Si3C |
-0.49 |
18.2 |
|
|
Fe82Cr2NbB8P4Si2C |
-0.42 |
4.06 |
|
|
Fe84Mo2B8Si2P4 |
-0.68 |
6.5 |
[8] |
|
Fe83Mo2B8Si2P4C1 |
-0.68 |
1.54 |
|
|
Fe84Cr2B8Si2P4 |
-0.69 |
0.68 |
|
|
Fe83Cr2B8Si2P4C1 |
-0.67 |
2.05 |
|
|
FeCrMoNiB |
-0.262 |
1.89 |
|
|
Fe83.3B16Cu0.7 |
-0.743 |
30 |
[9] |
|
Fe83.3Si1B15Cu0.7 |
-0.738 |
16 |
|
|
Fe83.3Si2B14Cu0.7 |
-0.737 |
22 |
|
|
Fe83.3Si3B13Cu0.7 |
-0.736 |
29 |
|
|
Fe72Ni8Si10B10 |
-0.501 |
15.4 |
[10] |
|
FeNiB |
-0.522 |
13 |
[11] |
|
FeNiB+6%Cr |
-0.394 |
4.4 |
|
|
FeNiB+11% Cr |
-0.272 |
0.29 |
|
|
FeNiB+17% Cr |
-0.256 |
0.15 |
|
|
FeNiB+26% Cr |
-0.264 |
0.15 |
|
|
FeNiB+33% Cr |
-0.256 |
0.14 |
|
|
Fe48Cr23Mo10C14B5 |
-0.859 |
3.03 |
[12] |
|
Fe68Cr8Mo4Nb4B16 |
-0.145 |
0.75 |
[13] |
|
Fe60Cr8Nb8B24 |
-0.23 |
0.41 |
[14] |
|
Fe75.8Si12B8Nb2.6Cu0.6P1 |
-0.579 |
17.4 |
[15] |
|
Fe38Co38Mo8Cu1B15 |
-0.597 |
8.21 |
[16] |
|
Fe39.5Co39.5Mo8Cu1B12 |
-0.691 |
20.2 |
|
|
Fe42Co42Cu1B15 |
-0.658 |
12.9 |
|
|
Fe63Co21Cu1B15 |
-0.723 |
33.1 |
|
|
Fe84Cu1B15 |
-0.874 |
36 |
|
|
Fe85B15 |
-1.010 |
64.7 |
|
|
Fe-Cr-Mo-C-B |
-0.799 |
145 |
[17] |
|
Fe-Cr-P-B-C |
-0.524 |
3.2 |
|
|
Fe-Cr-Mo-Ni-B-Co-Cu-Si-C |
-0.56 |
3.92 |
|
|
Fe-Cr-Mo-Ni-P-B-C-Si |
-0.45 |
8.3 |
|
|
Fe-Cr-Mo-C-B-Y |
-0.559 |
4.12 |
|
|
Fe-Cr-Mo-C-B |
-0.546 |
10 |
|
|
Fe-C-Si-B-P-Cr-Mo-Al |
-0.707 |
2.3 |
|
|
Fe-Cr-Mo-C-B-Y |
-0.458 |
11.3 |
|
|
Fe-Cr-Si-B-C |
-0.641 |
31.85 |
|
|
Fe-Cr-P-B-C |
-0.647 |
4.3 |
|
|
Fe-Cr-Mo-C-B-Y |
-0.679 |
5.091 |
|
|
Fe-Mo-Cr-Co |
-0.438 |
6.9 |
|
|
Fe-Cr-Mo-B-C |
-0.333 |
52.2 |
|
|
Fe-Cr-Mo-B-C-P |
-0.415 |
1.1 |
|
Table S1 The comparison data of corrosion potential (Ecorr) and corrosion current density (icorr) of various Fe-based alloys in 3.5 wt.% NaCl solution.
References:
- Hua, Y.; Mohammed, S.; Barker, R.; Neville, A., Comparisons of corrosion behaviour for X65 and low Cr steels in high pressure CO2-saturated brine. Journal of Materials Science & Technology 2020,41, 21-32.
- Lekakh, S. N.; Buchely, M.; Li, M.; Godlewski, L., Effect of Cr and Ni concentrations on resilience of cast Nb-alloyed heat resistant austenitic steels at extreme high temperatures. Materials Science and Engineering: A 2023,873, 145027.
- Wang, Y.; Yu, H.; Wang, L.; Li, M.; Si, R.; Sun, D., Research on the structure-activity relationship of Fe-Cr alloy in marine environment based on synchrotron radiation: Effect of Cr content. Surfaces and Interfaces 2021,26, 101370.
- Xiang, S.; Fan, Z.; Chen, T.; Lian, X.; Guo, Y., Microstructure evolution and creep behavior of nitrogen-bearing austenitic Fe–Cr–Ni heat-resistant alloys with various carbon contents. Journal of Materials Research and Technology 2023,23, 316-330.
- Zhang, C.; Li, Q.; Xie, L.; Zhang, G.; Mu, B.; Chang, C.; Li, H.; Ma, X., Development of novel Fe-based bulk metallic glasses with excellent wear and corrosion resistance by adjusting the Cr and Mo contents. Intermetallics 2023,153, 107801.
- Botta, W. J.; Berger, J. E.; Kiminami, C. S.; Roche, V.; Nogueira, R. P.; Bolfarini, C., Corrosion resistance of Fe-based amorphous alloys. Journal of Alloys and Compounds 2014,586, S105-S110.
- Han, Y.; Chang, C. T.; Zhu, S. L.; Inoue, A.; Louzguine-Luzgin, D. V.; Shalaan, E.; Al-Marzouki, F., Fe-based soft magnetic amorphous alloys with high saturation magnetization above 1.5 T and high corrosion resistance. Intermetallics 2014,54, 169-175.
- Han, Y.; Kong, F. L.; Han, F. F.; Inoue, A.; Zhu, S. L.; Shalaan, E.; Al-Marzouki, F., New Fe-based soft magnetic amorphous alloys with high saturation magnetization and good corrosion resistance for dust core application. Intermetallics 2016,76, 18-25.
- Fan, Y.; Zhang, S.; Xu, X.; Miao, J.; Zhang, W.; Wang, T.; Chen, C.; Wei, R.; Li, F., Effect of the substitution of Si for B on thermal stability, magnetic properties and corrosion resistance in novel Fe-rich amorphous soft magnetic alloy. Intermetallics 2021,138, 107306.
- Vasić, M. M.; Simatović, I. S.; Radović, L.; Minić, D. M., Influence of microstructure of composite amorphous/nanocrystalline Fe72Ni8Si10B10 alloy on the corrosion behavior in various environments. Corrosion Science 2022,204, 110403.
- Berger, J. E.; Jorge, A. M.; Koga, G. Y.; Roche, V.; Kiminami, C. S.; Bolfarini, C.; Botta, W. J., Influence of chromium concentration and partial crystallization on the corrosion resistance of FeCrNiB amorphous alloys. Materials Characterization 2021,179, 111369.
- Wu, L.; Zhou, Z.; Zhang, K.; Zhang, X.; Wang, G., Electrochemical and passive film evaluation on the corrosion resistance variation of Fe-based amorphous coating affected by high temperature. Journal of Non-Crystalline Solids 2022,597, 121892.
- Coimbrão, D. D.; Zepon, G.; Koga, G. Y.; Godoy Pérez, D. A.; Paes de Almeida, F. H.; Roche, V.; Lepretre, J. C.; Jorge, A. M.; Kiminami, C. S.; Bolfarini, C.; Inoue, A.; Botta, W. J., Corrosion properties of amorphous, partially, and fully crystallized Fe68Cr8Mo4Nb4B16 alloy. Journal of Alloys and Compounds 2020,826, 154123.
- Koga, G. Y.; Nogueira, R. P.; Roche, V.; Yavari, A. R.; Melle, A. K.; Gallego, J.; Bolfarini, C.; Kiminami, C. S.; Botta, W. J., Corrosion properties of Fe–Cr–Nb–B amorphous alloys and coatings. Surface and Coatings Technology 2014,254, 238-243.
- Liu, Y.; Li, J.; Sun, Y.; He, A.; Dong, Y.; Wang, Y., Effect of annealing temperature on magnetic properties and corrosion resistance of Fe75.8Si12B8Nb2.6Cu0.6P1 alloy. Journal of Materials Research and Technology 2021,15, 3880-3894.
- Sunbul, S. E.; Akyol, S.; Onal, S.; Ozturk, S.; Sozeri, H.; Icin, K., Effect of Co, Cu, and Mo alloying metals on electrochemical and magnetic properties of Fe-B alloy. Journal of Alloys and Compounds 2023,947, 169652.
- Meghwal, A.; Pinches, S.; King, H. J.; Schulz, C.; Stanford, N.; Hall, C.; Berndt, C. C.; Ang, A. S. M., Fe-based amorphous coating for high-temperature wear, marine and low pH environments. Materialia 2022,25, 101549.

Reviewer 2 Report
In this study, the role of Cr element in highly dense passivation of Iron-based amorphous alloy was investigated with electrochemical technique, nano intention, and various microstructure test methods. It is reported that Cr content affects corrosion resistance. The topic is interesting and has important applications for the steel industry.
The abstract and introduction are fairly explained. Please introduce a paragraph in the introduction that emphasizes the novelty of the present work. Please highlight the significance of the study. Elaborate why this study is important because corrosion nature in the presence of Cr has been studied many times even with the altering the concentration/composition. What aspect makes this study unique to attract a larger audience?
Few spelling mistakes in abstract e.g. - “nano indention” correct it nanoindentation and “ vairous” as various. I am not pointing out all the mistakes please proofread the manuscript.
Cr is somewhere small or subscript, Please clarify.
Please mention the factors affecting the corrosion behavior. e.g., high temperatures because it is one of the important aspects to consider. Please address this…
Could you please mention the planes in XRD?
Did you see any phase transformation? please explain this factor as numerous precipitates have been reported while changing Cr concentration.
The result and discussion portion is well explained and useful findings.
I would recommend citing some references for improving the literature part.
https://www.sciencedirect.com/science/article/abs/pii/S0968432819303713
https://www.sciencedirect.com/science/article/pii/S1369702108702050
There are a few spelling mistakes and vague sentences.
Author Response
Response to the comments on “Role of Cr element in highly dense passivation of Fe-based amorphous alloy”
- Please introduce a paragraph in the introduction that emphasizes the novelty of the present work. Please highlight the significance of the study. Elaborate why this study is important because corrosion nature in the presence of Cr has been studied many times even with the altering the concentration/composition. What aspect makes this study unique to attract a larger audience?
Re:
At present, a large number of parts fail due to corrosion in industrial production, especially in offshore oil and gas exploitation, so it is necessary to find an alloy with lower cost and better corrosion resistance. In this work, FeSiB system with better amorphous formation ability is selected and corrosion resistance elements Nb and Cr are added to replace Mo, which is relatively high cost in traditional Fe-Cr alloy, we hope to find an amorphous alloy with excellent properties, and explore the role of Cr in amorphous passivation film during corrosion, and the influence of Cr content on amorphous formation ability.
- Few spelling mistakes in abstract e.g. - “nano indention” correct it nanoindentation and “ vairous” as various. I am not pointing out all the mistakes please proofread the manuscript.
Re:
Thanks for this comment, we have checked and rectified the typos in the manuscript carefully.
- Cr is somewhere small or subscript, Please clarify.
Re:
We have corrected the content of the manuscript, only when representing the concentration of Cr is the subscript.
- Please mention the factors affecting the corrosion behavior. e.g., high temperatures because it is one of the important aspects to consider. Please address this…
Re:
There are many factors that affect corrosion behavior, such as temperature, impurities and oxidation. In this work, all electrochemical experiments were completed at room temperature (298K). At the same time, we added two references [13, 14] to introduce some new researches on the corrosion resistance of high Cr alloys at high temperatures were introduced in the introduction of the paper (Page 1, line 36-39 in the revised manuscript).
- Could you please mention the planes in XRD?
Re:
By comparing in Jade, we have labeled the peaks in the revised manuscript, as shown in Figure 1.
Figure 1. XRD curves of Fe72-xCrxB19.2Si4.8Nb4 ingots. (e) x = 0 (Cr0), (f) x = 7.2 (Cr7), (g) x = 21.6 (Cr21), (h) x = 36 (Cr36).
- Did you see any phase transformation? please explain this factor as numerous precipitates have been reported while changing Cr concentration.
Re:
From Figures 1h, 2d and 2e, it can be seen that Nb0.81Si0.19 disappeared during the rapid cooling process, but CrFeB and α-Fe remained, which also indicates that the precipitation of Cr-containing phase will occur when the cCr a certain content.
- I would recommend citing some references for improving the literature part
Re:
Thanks for this comment. In introduction, we have newly cited five relevant studies in the past three years [13-17], which are about the corrosion resistance of high Cr alloys in high-temperature, the threshold value of Cr content in Fe-based amorphous passivation film and the application of Marine corrosion, all of these are marked with red underlines in references (Page 15, line 569-573 and line 596-601 in the revised manuscript).
References:
- Lekakh, S. N.; Buchely, M.; Li, M.; Godlewski, L., Effect of Cr and Ni concentrations on resilience of cast Nb-alloyed heat resistant austenitic steels at extreme high temperatures. Materials Science and Engineering: A 2023,873, 145027.
- Xiang, S.; Fan, Z.; Chen, T.; Lian, X.; Guo, Y., Microstructure evolution and creep behavior of nitrogen-bearing austenitic Fe–Cr–Ni heat-resistant alloys with various carbon contents. Journal of Materials Research and Technology 2023,23, 316-330.
- Hua, Y.; Mohammed, S.; Barker, R.; Neville, A., Comparisons of corrosion behaviour for X65 and low Cr steels in high pressure CO2-saturated brine. Journal of Materials Science & Technology 2020,41, 21-32.
- Wang, Y.; Yu, H.; Wang, L.; Li, M.; Si, R.; Sun, D., Research on the structure-activity relationship of Fe-Cr alloy in marine environment based on synchrotron radiation: Effect of Cr content. Surfaces and Interfaces 2021,26, 101370.
- Zhang, C.; Li, Q.; Xie, L.; Zhang, G.; Mu, B.; Chang, C.; Li, H.; Ma, X., Development of novel Fe-based bulk metallic glasses with excellent wear and corrosion resistance by adjusting the Cr and Mo contents. Intermetallics 2023,153, 107801.

Reviewer 3 Report
The manuscript discusses the role of Cr element in highly dense passivation of Fe-based amorphous alloy. The investigated alloy is of high interest in several industrial and medical applications. The used methods are adequate and were well descibed in the text. However, minor changes are required to improve the quality of the manuscript. Furthermore, the conclusions section must be improved. Attached authors will find list of comments incorporated in the PDF file.

The English language requires improvement in many occasions as indicated in the attached PDF filde.
Author Response
Response to the comments on “Role of Cr element in highly dense passivation of Fe-based amorphous alloy”
- Minor changes are required to improve the quality of the manuscript. Furthermore, the conclusions section must be improved.
Re:
Thank you for all your comments in the PDF, we have carefully read and corrected the content of the manuscript. According to the advice, we moved the figure below the corresponding text, the first occurrence of the abbreviation is also given the full name and supplemented the conclusion section, which is marked with red underlines (Page 14, line 516-520 in the revised manuscript).
- Is Fe72-xCrxB2Si4.8Nb4(x = 0, 7.2, 21.6, 36) a nominal or actual composition? this is very important to mention.
Re:
Fe72-xCrxB19.2Si4.8Nb4 (x = 0, 7.2, 21.6, 36) is a nominal composition, we have revised the relevant content in the manuscript and marked with red underlines (Page 2, line 90-91 in the revised manuscript).
- What are the units used for these variables in formula (1)?
Re:
The unit of m0 and m1 is g, the unit of S is cm2, and the unit of t is h, which we have revised in the manuscript and marked with red underlines (Page 3, line 122-124 in the revised manuscript).
- It would be even better if you could find evidence for larger radius Cr atoms replacing Fe atoms
Re:
It can be seen from the XRD patterns that the θ of the three strong peaks has a decreasing trend. According to the Bragg equation (1), it is believed that d has gradually increases, therefore we thought that the addition of Cr atoms replace parts of Fe atoms.
2dsinθ=nλ (1)

Round 2
Reviewer 1 Report
Dear authors, eventhough you have revised the technical part as per my suggestions, but still, the similarity index is 32% and it is very much in the introduction, experimental and few places in the results. Authors have not taken any efforts in reducing the similarity.
Author Response
Response to the comments on “Role of Cr element in highly dense passivation of Fe-based amorphous alloy”
The similarity index is 32% and it is very much in the introduction, experimental and few places in the results. Authors have not taken any efforts in reducing the similarity.
Re:
We are very sorry for such problems in the revised manuscript last time. After discussion, we have revised the introduction, experiment and part of the results and discussion, which is marked with red underlines.

Reviewer 2 Report
The paper is acceptable in its current form.
Author Response
Re:
Thanks for your comments.